# Effect of Milling Parameters on Amplitude Spectrum of Vibrations during Milling Materials Based on Wood

**Áron Hortobágyi [1],\*, Peter Koleda [1], Pavol Koleda [1] and Richard Kminiak [2]**

[1] Department of Manufacturing and Automation Technology, Faculty of Technology, Technical University in Zvolen, T. G. Masaryka 24, 960 01 Zvolen, Slovakia

[2] Department of Woodworking, Faculty of Wood Sciences and Technology, Technical University in Zvolen, T. G. Masaryka 24, 960 01 Zvolen, Slovakia

\* Correspondence: xhortobagyi@is.tuzvo.sk

**Abstract:** Milling with use of CNC machines is a well-established method and much research was concluded on this topic. However, when it comes to wood and wood composites, the material non-homogeneity brings a lot of variability into cutting conditions. As a part of research into potential signals for nesting milling, material vibrations at clamping points were examined in this study. The main goal was to conclude if cutting parameters have a statistically significant effect on measurement. The place of measurement was analyzed so it was accessible to the machine operator. Medium density fiberboard and particleboard specimens were cut through by razor and spiral mill, with spindle rotating 10,000 and 20,000 $min^{-1}$ and feed rates 2, 6, 10 m·$min^{-1}$. Vibrations were measured at vacuum grippers, and were then processed by fast Fourier transform. Then, frequency spectrum maxima were compared, as well as amplitude sizes. Main frequencies were of roughly 166 Hz and multiples, suggesting their origin in tool rotation. When maxima were compared, tool use, spindle rotation, and feed rate seemed to affect the result. Frequency spectrum amplitudes were subjected to analysis of variance, significant effect was found on spindle speed, tool, and specimen material. No significant effect was found with differing feed rates.

**Keywords:** MDF; CNC; milling; vibration measurement





## 1. Introduction

The technology of machining native wood and wood-based materials by multi-axis CNC machining centers is increasingly used especially in the manufacturing of complex parts, or so-called nesting milling. CNC machines are often used without direct control by a human operator and, therefore, the setting of appropriate technological parameters is extremely important for trouble-free machining, achieving the required quality of the workpiece, minimizing vibration and electricity consumption, and, last but not least, ensuring adequate work environment [1,2]. The ongoing digital revolution (Industry 4.0) brings techniques for inspection and for collecting data on the production process and their online processing remotely using interconnected cyber-physical systems. It does not have to be only about evaluating data in real time, but also about their prediction with the aim of environmental protection and sustainability industry [3]. The problems related to this can result in the development of new intelligent sensors, methods, and procedures for measuring quantities that indirectly characterize the machining process (energy consumption, acoustic emissions, vibrations, dust) and are suitable for creating a digital twin of intelligent production [4].

Milling is a well-known manufacturing process, where many problems already have been addressed. Many studies were conducted to find dependence of roughness of machined material on cutting parameters. Generally, the surface roughness depends on spindle speed, feed rate, and tool diameter. To achieve a smoother surface, high spindle

speed with slow federate should be used, ideally with a small diameter tool [5–13]. However, such parameters also create higher cutting forces which are undesirable [14]. A big aspect is from material itself. Especially where wood is concerned, material homogeneity plays a big role.

Medium density fiberboard (MDF) is a wood-based industrial product. It is made from wood waste fibers bonded together by resin, while heated under pressure. MDF has certain advantages when compared to native wood and is currently preferred in many applications [10]. MDF roughness from manufacture increases with rising compression strength [15]. It is generally denser than particle board as well as plywood. Even though it consists of fibers and not veneers, it can be applied as a construction material in most cases, where plywood is used nowadays.

The density of a typical MDF is between 500 $\text{kg} \cdot \text{m}^{-3}$ and 1000 $\text{kg} \cdot \text{m}^{-3}$ whereas the density of a particle board is in the range from 160 $\text{kg} \cdot \text{m}^{-3}$ to 450 $\text{kg} \cdot \text{m}^{-3}$. In contrast to natural wood, MDF does not contain knots or rings [16]. Experimental studies provided assessment of the concrete cutting parameters for machining isotropic and orthotropic wood-based materials [17,18].

Vibrations during the machining process were subjected to multiple studies. Vibrations are often frequent problem which affects dimensional accuracy of the parts being machined, surface finish quality and tool life. Vibrations are induced due to machine faults, cutting tool, cutting parameters, workpiece deformation, etc. These vibrations are generally measured using accelerometers mounted on various machine parts elements [19].

From the point of view of vibration measurement, several articles were published, especially in the area of metal materials machining. A relation was found for steel [20] and titanium [21] machining, that surface roughness is most dependent on feed rate, while cutting speed has the highest effect on tool vibration. Similar results were found for aluminum in [22]. Other studies were focused on finding a relation between vibrations and surface finish, with signal spectrum analysis and wavelet packet transform (WPT), where it resulted in vibration ranges correlating the vibration amplitudes with resultant surface roughness. The measured vibration and wavelet packet transform method could be effectively applied for real-time, highly accurate, and reliable roughness monitoring, with a low computation power cost in CNC machining [23]. Singular spectrum analysis was also considered as a viable strategy for assessing vibration signals used to the real-time monitoring of machined surface finish [24], as well as the calculation of surface quality in CNC turning by model-assisted response surface approach [25]. Tool vibration signals were experimentally monitored by spectral kurtosis and ICEEMDAN energy modes for insert abrasion assessment [26]. Surface quality prediction models based on a regression method and artificial neural network were developed in [27,28]. Multiple methods involving machine learning were summarized and compared in [29]. Tool geometry was also found to be a big factor, when vibrations are assessed [30–32].

The fast Fourier transform used in this study is a valuable tool in situations when signal processing is needed. It is a computation instrument for easier signal analysis. FFT can be used on computers for power spectrum analysis as well as for filter simulation. This tool is basically an efficient way to calculate the discrete Fourier transform of data sample sequences [33] and to transform the time domain signal into frequency domain [34]. The Fourier transform was applied for surface roughness prediction to obtain the features of image texture [35], and for real-time measurement and intervention system for build-up-edge and tool damage to analyze the vibration signals for fast recognition of signal irregularities [36]. It was also successfully tested for on-the-fly CNC interpolation method [37]. In this research, fast Fourier transform was used to find dominant frequencies which were combined with noise in a composite signal. When multiple frequencies are combined, it is almost impossible to decompose a signal into original elements by signal shape assessment. After fast Fourier transform, two-sided amplitude spectrum and single-sided amplitude spectrum can be calculated with use of signal length. Other examples of the fast Fourier

transform use are the conversion of Gaussian pulse or the turning of periodical waves from the time domain to the frequency domain [38].

Experiments with MDF routing showed that a suitable cutting edge angle, with consideration of tool material, is in lower scope of ordinarily used HM tools. From the range of the research, the best performing angle was roughly 40 degrees [39]. Another factor that was both cause and effect of vibration is tool wear [2]. To mitigate this effect, many methods were tested, with a range of them focusing on MDF routing. An adaptive regulation system generating responses to advancing tool wear was developed in [40]. A neural network was tested for tool wear monitoring, where several machining factors were measured, including cutting forces, temperature, and power [41]. Mathematical models were also used for the correction of size parameters at 13 levels of a tool wear, with resulting control charts [42].

Another factor affecting vibration is clamping. Vacuum clamping systems, such as the one used in this study, are usually used for particle boards milling on woodworking machining centers. The vacuum gripper systems provide good access to the workpiece edges during machining. A downside of this securing method is un-clamped board areas with relative distance to nearest gripper. These are relatively free to vibrate in a wide frequency range while machining takes place. Due to these vibrations, the roughness of the machined edge is higher, and the process is accompanied by high acoustic emission [43]. However, all the above could be measured by other non-intrusive methods, such as vibro-acoustic analysis [44–46], or by means of energy consumption tests [47,48].

This research focused on the evaluation of vibrations during milling of medium density fiberboards on 5-axis CNC machine. The goal of the study was a confirmation of the effect of changing the technological parameters of milling wood-based agglomerates on the size of the vibration amplitude. Vibration measurement should serve as one of the appropriate signals for an adaptive machining control system, which is the goal of ongoing research of FMA analysis of potential signals suitable for adaptive control of nesting strategies for milling wood-based agglomerates. The research was intended to confirm the hypotheses that a change in spindle speed, feed rate, workpiece material, and a change in the tool influences the change in the amplitude of the resulting vibrations. At the same time, one of the goals of the article was to measure these vibrations in a place accessible to the operator.

## 2. Materials and Methods

Medium density fiberboards cut to $500 \times 300 \times 18$ mm with weight 1960 g were used as specimens. The density of these specimens was 720–740 kg·m$^{-3}$. Particleboards of the same dimensions were added as reference specimens. The density of particleboards given by the manufacturer was 600–640 kg·m$^{-3}$ (deciduous 10%, coniferous 90%), and urea formaldehyde glue with paraffin admixture was used; both originated from Kronospan Ltd., Zvolen, Slovakia. The manufacturer declared that the material complied with the EN 14,322 standard, EN 312-2, and emission class E1 (EN ISO 12460-5) [49–51].

A measurement was conducted on a 5-axis CNC machining center SCM Tech Z5 (Figure 1), in laboratories of Technical University in Zvolen. Table 1 provides the basic technical-technological parameters given by the manufacturer.

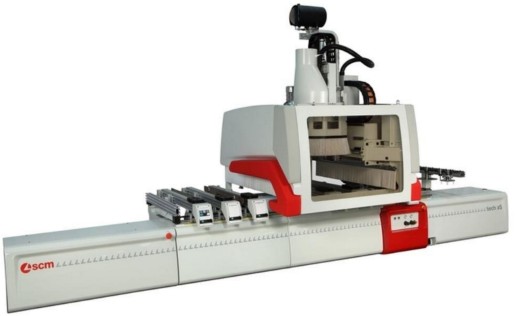

**Figure 1.** CNC machining center SCM Tech Z5.

**Table 1.** CNC machining center SCM Tech Z5 technical parameters.

| CNC Machining Center SCM Tech Z5 Technical Parameters | |
|---|---|
| Useful desktop (mm) | X = 3050, Y = 1300, Z = 3000 |
| Speed in $x$ axis (m·min$^{-1}$) | $0 \div 70$ |
| Speed in $y$ axis (m·min$^{-1}$) | $0 \div 40$ |
| Speed in $z$ axis (m·min$^{-1}$) | $0 \div 15$ |
| Vector rate (m·min$^{-1}$) | $0 \div 83$ |
| **Technical Parameters of the Electric Spindle with HSK F63 Connection** | |
| Rotation in C axis | 640° |
| Rotation in B axis | 320° |
| Revolutions (min$^{-1}$) | $600 \div 24000$ |
| Electric power (kW) | 11 |
| Maximum tool dimensions (mm) | D = 160, L = 180 |

Vibration was measured by PicoScope with MEMS accelerometer TA143 [52] with parameters in Table 2.

**Table 2.** Basic parameters of accelerometer TA143.

| Parameter | Value |
|---|---|
| Maximum measurable acceleration | ±5 g |
| Output | 0–2 V DC |
| Output scaling | 99 to 122 mV·g$^{-1}$ |
| 0 g output | 0.85 to 1.15 V |

Specimens were clamped to four pneumatic grippers, each with surface $120 \times 120$ mm and clamping force 16 kg/m$^2$ (Figure 2). Each cut had a depth of 19 mm, and so, the whole thickness of the material was machined in one run.

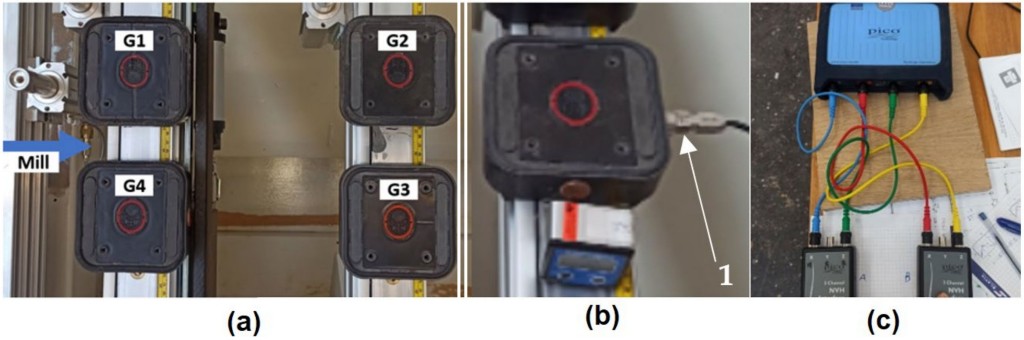

**Figure 2.** Measurement assembly. (**a**) Pneumatic grippers (G1–G4), (**b**) magnetic attachment of accelerometer (1), (**c**) NVH kit with PicoScope.

To emulate the conditions of nesting milling, a tool trajectory cut through the middle of specimen. The spiral and razor cutter shown in Figure 3 were used as tools, with parameters summed in Table 3.

Varying feed rate of 2, 6, 10 m·min$^{-1}$ was used, with spindle rotations 10,000 and 20,000 min$^{-1}$. Cutting was repeated 3 times for each parameter. Both experimental layout and cutting parameters are shown in Figure 4 and Table 4 [53].

As seen in Figures 2 and 4, accelerometer probes were attached onto pneumatic grippers. To select placement location, multiple tests were conducted with probes placed on each gripper. An example of a result with spindle revolutions of 20,000 min$^{-1}$ and feed rate of 6 m·min$^{-1}$ is shown in Figure 5.

Although cutting parameters were the same, the placement of accelerometer had a visible impact. Grippers G1 and G4 had a larger peak in the beginning of cutting, as they

were closer to the start of tool path. However, there was still a big difference, due to tool rotation. On gripper G1 and G2, vibrations were caused by conventional milling, while grippers G3 and G4 side were experiencing climb milling. Another useful fact was that vibrations in *y* axis never showed as maximum in measurement.

Therefore, it was concluded that four channels of accelerometer were used to simultaneously measure x and z axes of two grippers. Goal was to have at least one sensor as close to the source as possible, and so, a pair of grippers, one near start and one near end position, were chosen. Lastly, as grippers G3 and G4 showed larger extremes, they were the final choice and sensors were placed as shown in Figure 4.

Data were then processed in MATLAB (MathWorks, Inc., Sentic, MA, USA). The initial signal was cut, so only the milling part would be assessed, and measured voltage was converted to acceleration by output scaling [52].

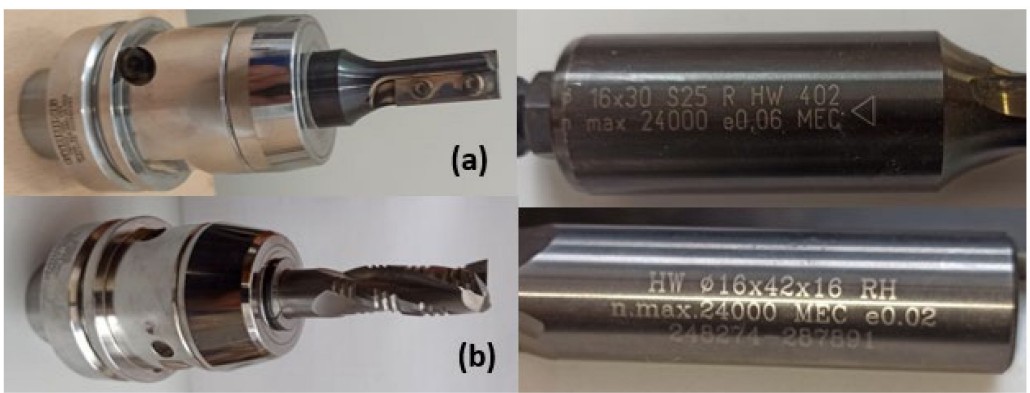

**Figure 3.** Cutters used in experiment. (**a**) Razor cutter, (**b**) spiral cutter.

**Table 3.** Tool parameters.

| Parameter | Razor Mill | Spiral Mill |
|---|---|---|
| Flute diameter | 16 mm | 16 mm |
| Shaft diameter | 25 mm | 16 mm |
| *n* max | 24,000 | 24,000 |
| Teeth | 2 | 3 |
| Cut direction | straight | up-cut |
| Cutting edge | IGM D16 L28.3 | solid |
| Chip breaker | - | yes |
| Tool carrier | HSK 63 GM 300 | HSK 63 GM 300 |
| Reduction sleeve | - | 16–25 |

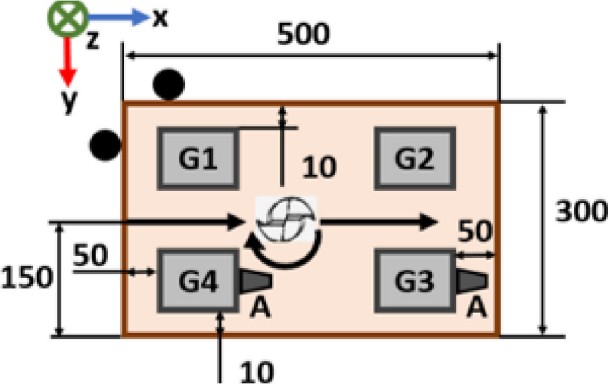

**Figure 4.** Experiment layout: Specimen placement and tool path on CNC. G—pneumatic gripper, A—accelerometer, x, y, z-orientation of accelerometer axes, G1–G4—Grippers no. 1–4.

**Table 4.** Number of measurements for each combination of cutting parameters.

| Cutting Parameters | | | VF [m·min⁻¹] | | | | | | | | |
| --- | --- | --- | --- | --- | --- | --- | --- | --- | --- | --- | --- |
| | | | 2 | | | 6 | | | 10 | | |
| Tool | $n$ [min⁻¹] | Grip. | x | y | z | x | y | z | x | y | z |
| spiral mill | 10,000 | 1 | - | - | - | - | - | - | - | - | - |
| | | 2 | - | - | - | - | - | - | - | - | - |
| | | 3 | 3 | - | 3 | 3 | - | 3 | 3 | - | 3 |
| | | 4 | 3 | - | 3 | 3 | - | 3 | 3 | - | 3 |
| | 20,000 | 1 | - | - | - | 3 | 3 | 3 | - | - | - |
| | | 2 | - | - | - | 3 | 3 | 3 | - | - | - |
| | | 3 | 3 | - | 3 | 3 | 3 | 3 | 3 | - | 3 |
| | | 4 | 3 | - | 3 | 3 | 3 | 3 | 3 | - | 3 |
| razor cutter | 10,000 | 1 | - | - | - | - | - | - | - | - | - |
| | | 2 | - | - | - | - | - | - | - | - | - |
| | | 3 | 3 | - | 3 | 3 | - | 3 | 3 | - | 3 |
| | | 4 | 3 | - | 3 | 3 | - | 3 | 3 | - | 3 |
| | 20,000 | 1 | - | - | - | - | - | - | - | - | - |
| | | 2 | - | - | - | - | - | - | - | - | - |
| | | 3 | 3 | - | 3 | 3 | - | 3 | 3 | - | 3 |
| | | 4 | 3 | - | 3 | 3 | - | 3 | 3 | - | 3 |

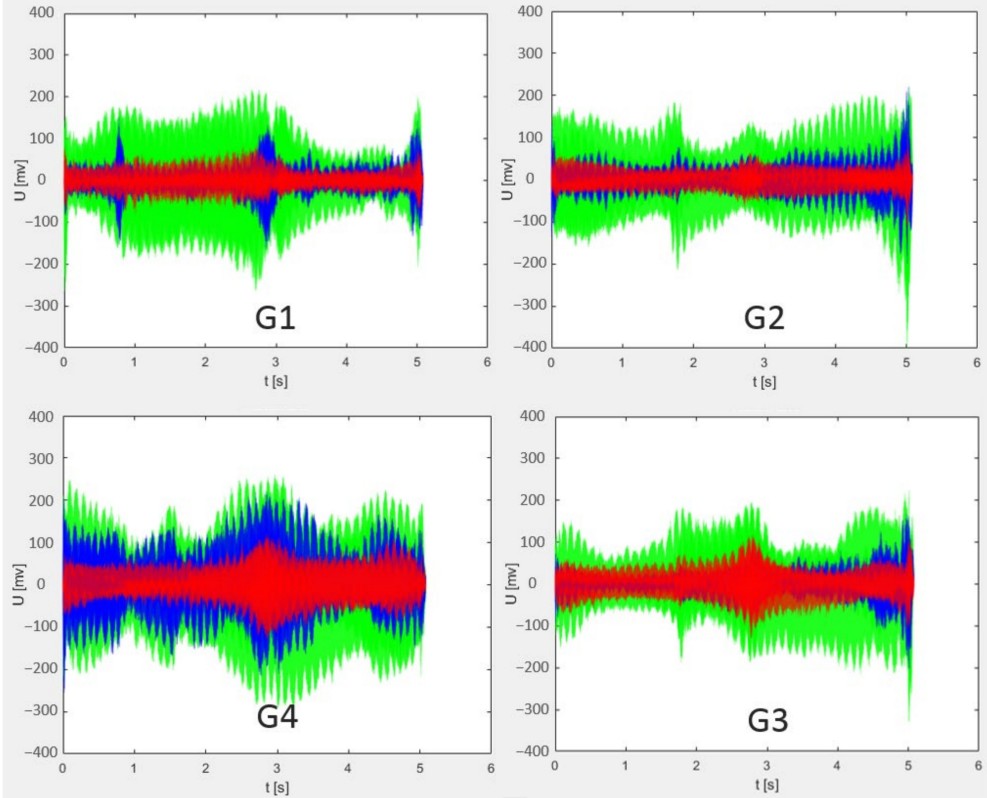

**Figure 5.** Result of measurement on individual axes. (**G1**–**G4**)—pneumatic grippers no. 1 to 4, blue—*x* axis, red—*y* axis, green—*z* axis.

Parts of the signal which contained information from before and after the milling process were cut. The threshold value ± 7 mV was used to determine the beginning and ending of the desired signal. Then, fast Fourier transform in MATLAB was used to calculate single-sided amplitude spectrum of vibrations [38] (percent sign "%" marks comments in MATLAB code):

```
L = length (X);
Fs = sampling frequency;
T = 1/Fs;
T = (0:L−1)*T;
Y = fft(X); % computing of FFT;
P2 = abs(Y/L);
P1 = P2(1:L/2 + 1);
P1(2:end−1) = 2*P1(2:end−1);
f = Fs*(0:(L/2))/L;
plot(f,P1); % generating of FFT graph;
title ("Single-Sided Amplitude Spectrum of S(t)"); % title of generated graph;
xlabel ("f (Hz)"); % x and y labels of generated graph;
ylabel(" | P1(f) | ");
```

In this code, X is input variable as sequence of acceleration according to the sampling period; L is number of samples in X; *Fs* is the value of sampling frequency; *T* is sampling period; t is time of discrete sample according to sampling period; Y is result of FFT from signal X; P2 are absolute values of Y/L ratio; P1 is computing of one-side spectrum amplitudes.

As there were multiple low peaks (noise) in mixture with significantly higher amplitude peaks, a threshold value of | Y(*f*) | = 0.5 was set as minimum for result to be recorded. Additionally, multiple frequency maxima were concentrated around certain values. For better readability, only the maximal value was recorded from such groups. Original signal, its cut form, and dominant frequencies with their maxima are shown in Figure 6.

The entire procedure of the experiment described in previous paragraphs is graphically displayed in Figure 7.

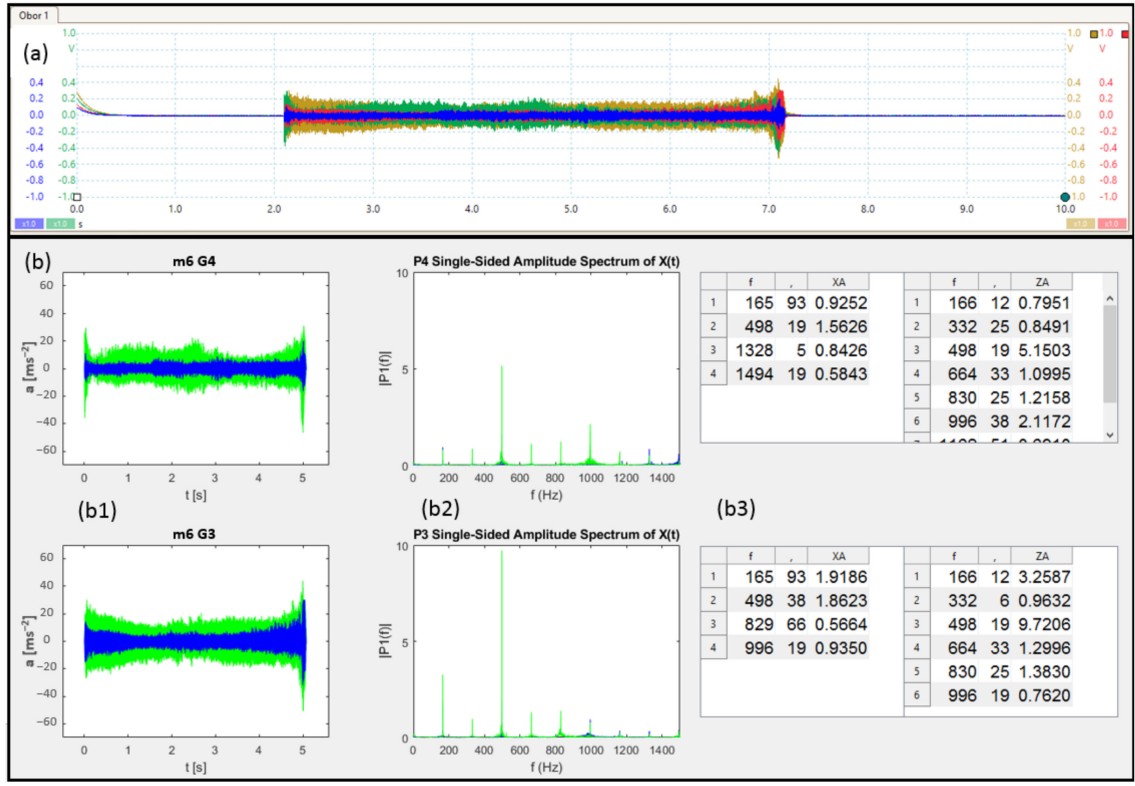

**Figure 6.** Example of signal processing. (**a**) PicoScope output showing vibrations of four channels, blue—*x* axis of gripper 4, green—*y* axis of gripper 4, red—*x* axis of gripper 3, yellow—*y* axis of gripper 3, (**b**) MATLAB output, (**b1**) signal from PicoScope (**b2**) Amplitude spectrum as FFT result (**b3**) list of peaks and their maxima, blue—*x* axis result, green—*z* axis result.

**Specimen preparation**
- Sawing of wood specimens with dimensions of 500 × 300 × 18 mm (length × width × thickness)
- Material of specimens: MDF and PTB

**Selection of milling parameters**
- $n$: 10,000 and 20,000 min$^{-1}$
- $v_f$: 2, 6, 10 m·min$^{-1}$
- tool: spiral and razor mill

**Milling of specimen at CNC machine**
- Measuring of vibrations at all four grippers
- Sellection of two grippers with highest amplitudes for further measurements

**Processing of measured signal**
- Conversion of voltage signal in [mV] into acceleration in [m·s$^{-2}$]
- Calculation of amplitude spectrum using FFT function in MatLab
- Filtering amplitude spectrum with treshold value of 0.5

**Statistical analysis of amplitudes**
- Analysis of variance of milling parameters effect on vibrations amplitudes
- Evaluation of probability of similarity of measured data at changed milling parameters using Duncan test

**Figure 7.** Scheme of experiment procedure.

## 3. Results

To enable data assessment, the results shown in Figure 6(b3) were also programmatically saved to tables, as shown in Table 5. Each dominant frequency is marked by bold font.

**Table 5.** Example of filtered output from fast Fourier transform.

| G3 X | | G3 Z | | G4 X | | G4 Z | |
|---|---|---|---|---|---|---|---|
| $f$ (Hz) | Maxima | $f$ (Hz) | Maxima | $f$ (Hz) | Maxima | $f$ (Hz) | Maxima |
| 165.93 | 0.93 | 166.13 | 0.80 | **165.93** | **1.92** | 166.13 | 3.26 |
| **498.20** | **1.56** | 332.26 | 0.85 | 498.39 | 1.86 | 332.06 | 0.96 |
| 1328.06 | 0.84 | **498.20** | **5.15** | 829.67 | 0.57 | **498.20** | **9.72** |
| 1494.19 | 0.58 | 664.33 | 1.10 | 996.19 | 0.94 | 664.33 | 1.30 |
| | | 830.26 | 1.22 | | | 830.26 | 1.38 |
| | | 996.39 | 2.12 | | | 996.19 | 0.76 |
| | | 1162.52 | 0.69 | | | | |
| | | 1328.65 | 0.51 | | | | |

Dominant frequencies were summarized. In most cases, when no parameters changed, they remained similar. A summary of their averages is shown in Figure 8.

Next, to determine if vibrations on the gripper could be used as a signal for adaptive control, the dependency between cutting parameters and single-sided amplitude spectrum amplitudes were assessed by analysis of variance (ANOVA). This analysis was conducted for significance level p = 5%. Figure 9 shows the dependency of amplitudes when changing spindle rotation from 10,000 to 20,000 min$^{-1}$. As the revolutions increased, the vibration amplitudes also increased.

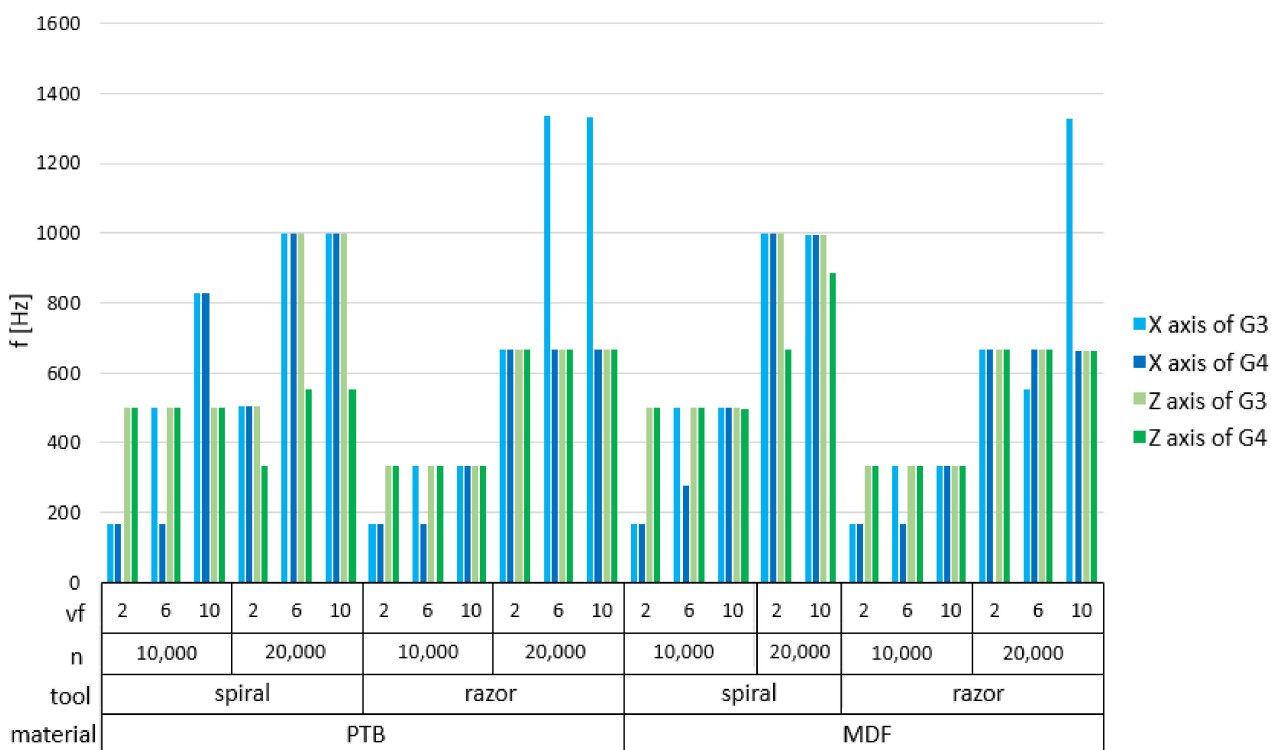

**Figure 8.** Summary of dominant frequency maxima though range of parameters.

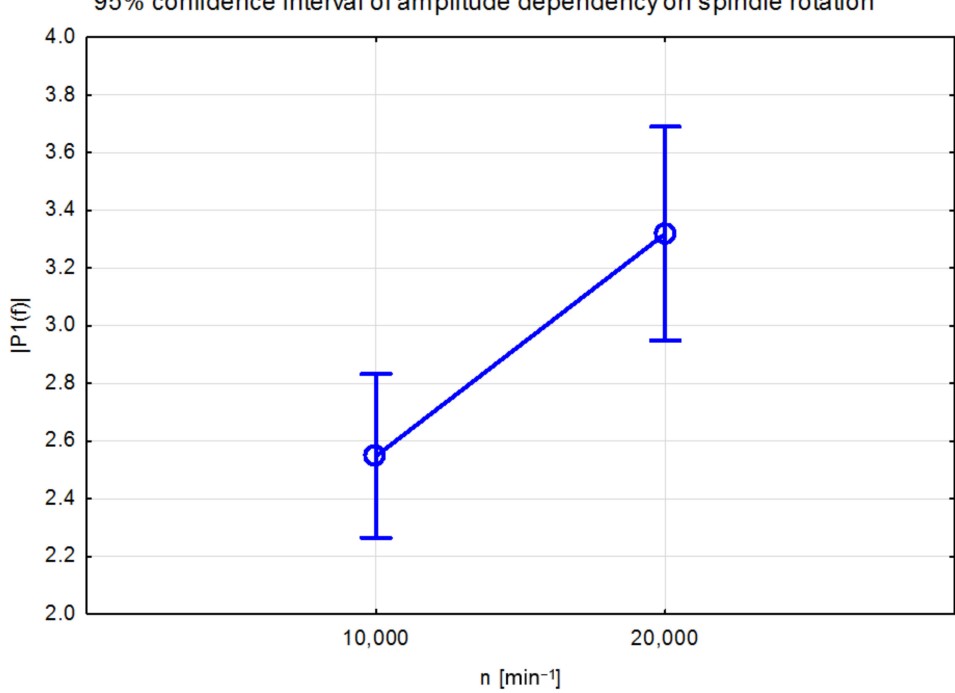

**Figure 9.** Box plot of 95% confidence interval of amplitude dependency on spindle rotation.

Figure 10 shows the dependency of amplitudes when changing the spindle rotations for different materials. When milling particleboard, amplitudes were lower than in milling MDF board.

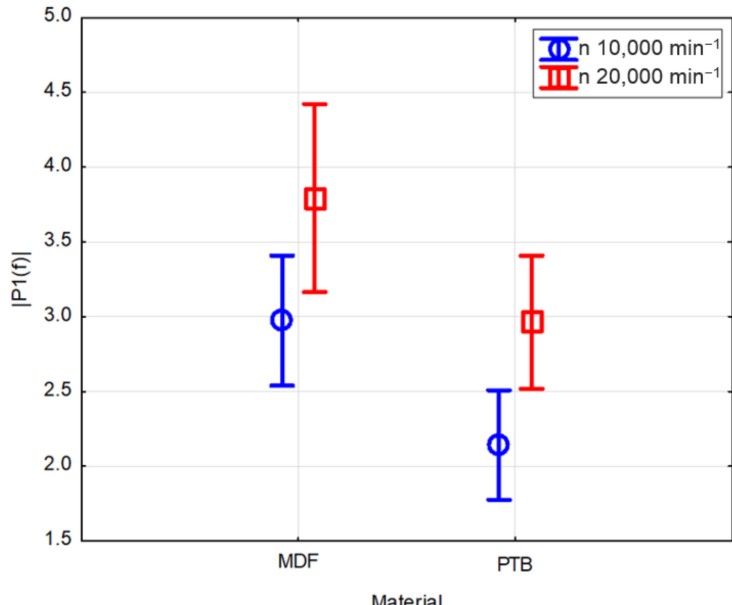

**Figure 10.** Box plot of 95% confidence interval of amplitude dependency on material and spindle rotation.

Figure 11 shows the dependency of amplitudes when changing feed rate for different experimental materials. As the feed rate increased, amplitudes increased at milling MDF boards bud decreased at milling particleboards.

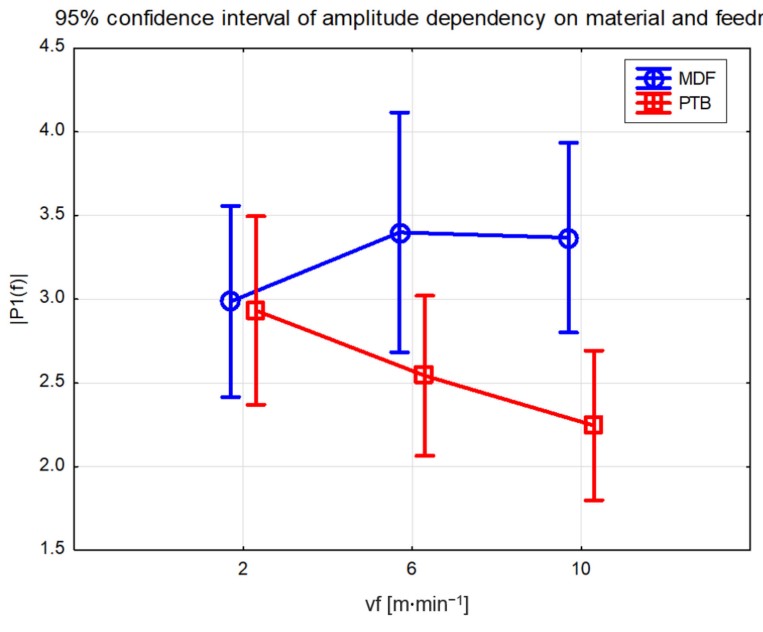

**Figure 11.** Box plot of 95% confidence interval of amplitude dependency on material and feed rate.

Figure 12 shows the dependency of amplitudes for different tools and experimental materials. The razor mill produced higher amplitudes of vibrations than the spiral mill at both experimental materials.

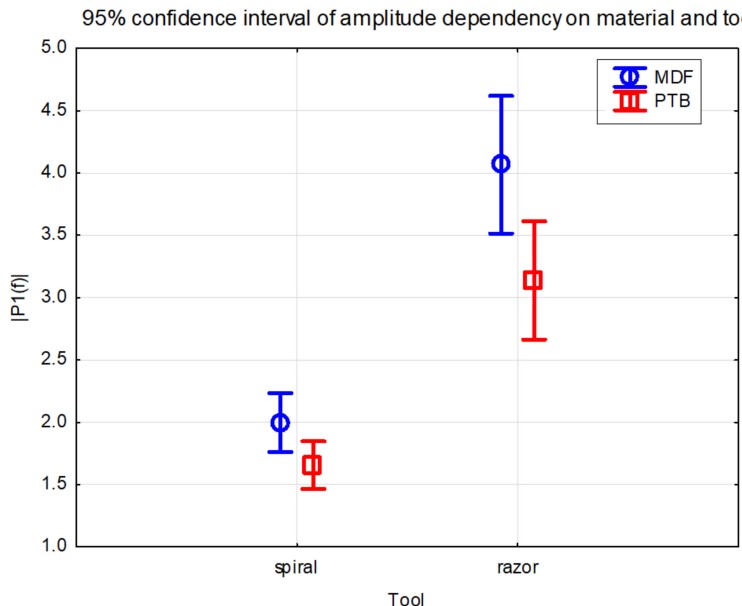

**Figure 12.** Box plot of 95% confidence interval of amplitude dependency on material and tool.

To verify the significance of differing parameters affecting amplitudes, the Duncan test was performed. A significant difference was found between tools, as shown in Table 6, spindle rotations are shown in Table 7, and materials are shown in Table 8. The change in feed rate, as shown in Table 9, did not impact amplitudes significantly.

**Table 6.** Duncan test of spindle rotation effect on amplitudes.

| No. | n [min$^{-1}$] | Ampl. Mean | 1 | 2 |
|---|---|---|---|---|
| 1 | 10,000 | 2.54 | | 0.000842 |
| 2 | 20,000 | 3.32 | 0.000842 | |

**Table 7.** Duncan test of tool effect on amplitudes.

| No. | Tool | Ampl. Mean | 1 | 2 |
|---|---|---|---|---|
| 1 | spiral | 1.80 | | 0.000009 |
| 2 | razor | 3.59 | 0.000009 | |

**Table 8.** Duncan test of material effect on amplitudes.

| No. | Material | Ampl. Mean | 1 | 2 |
|---|---|---|---|---|
| 1 | MDF | 3.28 | | 0.000540 |
| 2 | PTB | 2.49 | 0.000540 | |

**Table 9.** Duncan test of feed rate effect on amplitudes.

| No. | $v_f$ [m·min$^{-1}$] | Ampl. Mean | 1 | 2 | 3 |
|---|---|---|---|---|---|
| 1 | 2 | 2.96 | | 0.899389 | 0.490981 |
| 2 | 6 | 2.92 | 0.899389 | | 0.540870 |
| 3 | 10 | 2.75 | 0.490981 | 0.540870 | |

## 4. Discussion

The fast Fourier transform brought several results. First, as shown in Table 5, were dominant frequencies. The first peaks were observed at 165.93–166.13 Hz. The next peaks

were observed at 332.26 Hz and 498.2 Hz. These, with rest of peaks, presented multiples of the first value. In some cases, multiples are not seen in Table 5. However, these were still present, but with maxima smaller than 0.5;0 they were filtered out. Similar results were observed through use of all parameters. Therefore, it seems the main vibrations were caused by the rotating tool.

This theorem would be supported by the maxima summary shown in Figure 8. When maxima from spindle rotation $n = 10,000$ min$^{-1}$ are compared, the vibrations in $x$ axis seem smaller or equal to axis $z$. As expected, the spiral mill produced higher results mainly in the $z$ axis. With higher cutting speed $n = 20,000$ min$^{-1}$, the use of spiral mill produced larger maxima, which mostly seemed equally high in both $x$ and $z$ axis.

When comparing results from variance analysis, almost all parameter changes significantly affected the results. In Figure 9, the results show that the spindle speed had an effect on amplitude overall, with values in range (2.28–2.85) for $n = 10,000$ min$^{-1}$ and (2.95–3.7) for $n = 20,000$ min$^{-1}$. The significance of this result is shown in Table 6. When different material was also considered, as shown in Figure 10, the results were similar—amplitude ranges were higher with higher cutting speed. Ranges were overall smaller with MDF, when compared to PTB.

The effect of the varying feed rate, as shown in Figure 11, did not seem as clear. While it could be stated for particleboard that a higher feed rate lowered maximum amplitudes, the same cannot be deduced for MDF. The result reached for PTB also seemed to contradict research where a lower feed rate resulted in higher surface quality [7]. The uncertainty of this result can be also seen in the results of the post hoc test shown in Table 9. This might be due to varying density of boards, where a larger number of specimens would bring a more definite result.

Lastly, as shown in Figure 12, the tool impacts were compared. In both materials, the use of spiral mill caused smaller amplitude ranges—(1.75–2.25) for MDF and (1.45–1.8) for PTB. The use of razor mill resulted in higher ranges (3.5–4.6) for MDF and (2.7–3.6) for PTB. These results were also confirmed by the Duncan test, as shown in Table 7.

## 5. Conclusions

The main goal of our research was to determine whether vibration monitoring on a gripper could be used for adaptive control, and therefore, if changes during the milling process would be detectable. To answer this question, tests were conducted to find if changes of parameters would produce differing measurements. The fast Fourier transform was used to process measured signals.

The most dominant frequencies (133 Hz and higher multiples) seemed to originate from tool cutting into material. When single-sided spectrum maxima were compared, the tool, spindle rotation, and feed rate seemed to affect the result.

The variance analysis and Duncan tests revealed a significant effect of tool, material, and spindle rotations. The feed rate analysis did not show conclusive results where the probability of similarity was over 5%.

Overall, vibrations measured at pneumatic gripper processed with fast Fourier analysis seemed to be sufficient as a potential signal for adaptive control during milling, similarly to [22,23]. As in [23], the roughness of specimens should be measured and paired with vibrations in a further study, to provide data for adaptive control model.

The results and procedures of this study will also serve for measurements of amplitude spectrum during nesting milling strategies in ongoing research. The limitation of the investigated measuring and processing system was that if the signal is to be evaluated in real time, it will require higher computational demands. The authors also intend to use this system in the development of a smart pneumatic gripper for the woodworking industry, where the modified procedures described in the article can be used.

**Author Contributions:** Conceptualization, P.K. (Peter Koleda) and R.K.; methodology, P.K. (Peter Koleda); software, Á.H.; validation, P.K. (Peter Koleda); formal analysis, Á.H.; investigation, P.K. (Peter Koleda) and P.K. (Pavol Koleda); resources, R.K.; data curation, Á.H.; writing—original draft preparation, Á.H.; writing—review and editing, P.K. (Peter Koleda); visualization, Á.H.; supervision, P.K. (Peter Koleda); project administration, P.K. (Peter Koleda); funding acquisition, P.K. (Peter Koleda). All authors have read and agreed to the published version of the manuscript.

**Funding:** This research was funded by "APVV-20-0403 FMA analysis of potential signals suitable for adaptive control of nesting strategies for milling wood-based agglomerates" and project VEGA 1/0791/21 "Research of non-contact method of analysis of small and dust particles arising in the production process with a prediction of negative effects of dust particles".

**Institutional Review Board Statement:** Not applicable.

**Informed Consent Statement:** Not applicable.

**Data Availability Statement:** Not applicable.

**Acknowledgments:** The authors would like to thank the Operational Program Integrated Infrastructure for the project: National Infrastructure for Supporting Technology Transfer in Slovakia II-NITT SK II, co-financed by the European Regional Development Fund.

**Conflicts of Interest:** The authors declare no conflict of interest.

## Nomenclature

| | |
|---|---|
| CNC | Computer Numerical Control |
| DC | Direct Current |
| FFT | Fast Fourier Transform |
| FMA | Failure Mode Analysis |
| HM tools | Hard Metal tools |
| ICEEMDAN | Improved Complete Ensemble Empirical Mode Decomposition with Adaptive Noise |
| MDF | Medium-Density Fiberboard |
| MEMS | Micro Electronic Mechanic System |
| $N$ | revolutions ($\text{min}^{-1}$) |
| PTB | Particleboard |
| WPT | Wavelet Packet Transform |
| $v_f$ | feed rate ($\text{m}\cdot\text{min}^{-1}$) |

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
