# Peer review of "Effect of Milling Parameters on Amplitude Spectrum of Vibrations during Milling Materials Based on Wood"

_applsci, doi:10.3390/app13085061_

Round 1

Reviewer 1 Report

Please do not use the word "sample" in relation to the research material - the preferred word is "specimen". Please fix it in manuscript.

Nomenclature must be found in the paper. Full list of abbreviations, symbols and designations. Please add this to your manuscript even at the end of it.

It seems to me that abstract is too long - please shorten it, do not give numerical data. Abstract should be more condensed.

The authors must systematize the notation of the decimal place in numbers in the manuscript. Sometimes it's a period, sometimes it's a comma. There is no consistency between the text of the thesis and the tables. Please fix it in manuscript. If possible, the same convention should be maintained in the figures - but this largely depends on the software used by the authors to create subsequent graphs.

Table 3 seems to be misplaced - please check and correct it.

I suggest moving and enlarging the table in Figure 4b to make it more readable.

Figure 5 is illegible and shows very little. Graphs should be corrected and enlarged.

The notation of the formula marked as (1) is unclear. Please replace the "=" sign with something here. I suggest writing the form "=>". This formula also uses a dot instead of the product sign.

The MATLAB source code presented by the authors may mean nothing to the average reader who is not familiar with programming in MATLAB. This code should be enriched with a block diagram, add comment lines to it, what each of the instructions is for. Please also change the font for the code - I suggest "CONSOLAS". This code must be understandable. It is best if the authors also add a block diagram (another one) of their entire procedure in the experiment. The quantities appearing in the code should also be explained and described - precisely defined.

What is "maxima" in Table 4 - please clarify the explanation.

Please increase the font size in Figure 7 - especially in the description of both axes and the legend.

What are the bolded numbers in the following tables 5-8 in the first header rows, marked in red? For example, table 5. Please clarify this for the reader.

Please work on the quality of figures - the size of fonts, legends, descriptions of ordinate and abscissa axes must be systematized.

The paper has potential, but I have my doubts. I propose a minor / major revision, with an indication of the latter option. Please correct the manuscript and send it back for re-review.

Author Response

Please do not use the word „sample“ in relation to the research material – the preferred word is „specimen“. Please fix it in manuscript.

Authors answer: We have fixed this.

Nomenclature must be found in the paper. Full list of abbreviations, symbols and designations. Please add this to your manuscript even at the end of it.

Authors answer: We added nomenclature at the end of manuscript

It seems to me that abstract is too long – please shorten it, do not give numerical data. Abstract should be more condensed.

Authors answer: We have shortened the abstract.

The authors must systematize the notation of the decimal place in numbers in the manuscript. Sometimes it’s a period, sometimes it’s a comma. There is no consistency between the text of the thesis and the tables. Please fix it in manuscript. If possible, the same convention should be maintained in the figures – but this largely depends on the software used by the authors to create subsequent graphs.

Authors answer: We have fixed this as well as it was possible.

Table 3 seems to be misplaced – please check and correct it.
Authors answer: We have fixed this.

I suggest moving and enlarging the table in Figure 4b to make it more readable.

Authors answer: We have fixed this.

Figure 5 is illegible and shows very little. Graphs should be corrected and enlarged.
Author’s answer: We have fixed this.

The notation of the formula marked as (1) is unclear. Please replace the „=“ sign with something here. I suggest writing the form „=>“. This formula also uses a dot instead of the product sign.

Authors answer: This is only conversion relation. We have rewritten it into text in previous paragraph.

The MATLAB source code presented by the authors may mean nothing to the average reader who is not familiar with programming in MATLAB. This code should be enriched with a block diagram, add comment lines to it, what each of the instructions is for. Please also change the font for the code – I suggest „CONSOLAS“. This code must be understandable. It is best if the authors also add a block diagram (another one) of their entire procedure in the experiment. The quantities appearing in the code should also be explained and described – precisely defined.

Authors answer: Thanks for your comment. We added some comments and meaning of variables under this code.

What is „maxima“ in Table 4 – please clarify the explanation.
Authors answer: We have fixed this.

Please increase the font size in Figure 7 – especially in the description of both axes and the legend.

Authors answer: We have increased the font size.

What are the bolded numbers in the following tables 5-8 in the first header rows, marked in red? For example, table 5. Please clarify this for the reader.

Authors answer: We have corrected these tables so they could be more understandable for reader.

Please work on the quality of figures – the size of fonts, legends, descriptions of ordinate and abscissa axes must be systematized.

Authors answer: We did our best in revised paper.

Reviewer 2 Report

This paper studied the material vibrations at clamping points during CNC milling of wood. This work is interesting and has been well presented. The following questions should be revised before future publication.

1. Abstract: Improve the writing of the abstract. Firstly, the significance or starting point of the study should be briefly introduced at the beginning. Secondly, the sentence of “Main goal was to conclude if warrying cutting parameters have a measurable effect on measurement” is not clear enough. Thirdly, the unit of spindle rotating should be “r/min” or “rpm” rather than “m.min-1” and “min-1”. Moreover, in “significant effect was found on spindle speed (2,28 – 2,85) for n = 10 000 min-1 and (2,95 – 3,7) for n = 20 000 min-1” , the parameters should be given for  (2,28 – 2,85) and (2,95 – 3,7). Finally, the abbreviation of MDF and PTB should be changed to the full name when they first appeared.

2. Introduction: the writing of the current research review is fine. The research purpose is also well refined from the current literature review. However, some lately publications on vibration during machining are missed in the literature review. For instance, a paper on milling composites with spiral cutters (https://doi.org/10.1016/j.jmrt.2022.12.054) also reached the conclusion of  “Generally, the surface roughness decreased with increasing spindle speed…the increase of tool diameter [5-13]”. Another research (https://doi.org/10.1016/j.cja.2022.12.009) on titanium machining also mentioned the topic of “feed rate is the most dominating parameter affecting surface finish, whereas cutting speed is the major factor effecting tool vibration [20]”, which can be added to enrich your literature review.

4. Table 1: what is the meaning of unit of “A÷B” ?  Does it mean from A to B?

5. Improve the format of Fig 2, 3 and Table 3. Figures a and b need to have clear boundaries and cannot be mixed. Magnetic component should be pointed out in Fig. 2b.

6. The material of cutter should be given.

7. The labels should be given for the result of measurement on individual axes in Fig. 5.

8. Section 2: the units in “Varying feed rate of 2, 6, 10 m.s -1 was used, with spindle rotations 10 000 and 20 000 s -1” should be re-checked.

9. Fig. 9, 10, 11: the label of “n 10 000 min -1,  n 20 000 min -1” should be moved to the blank position in the upper right corner in the figure.

10. Improve the conclusion.

Author Response

Abstract: Improve the writing of the abstract. Firstly, the significance or starting point of the study should be briefly introduced at the beginning. Secondly, the sentence of “Main goal was to conclude if warrying cutting parameters have a measurable effect on measurement” is not clear enough. Thirdly, the unit of spindle rotating should be “r/min” or “rpm” rather than “m.min-1” and “min-1”. Moreover, in “significant effect was found on spindle speed (2,28 – 2,85) for n = 10 000 min-1 and (2,95 – 3,7) for n = 20 000 min-1” , the parameters should be given for  (2,28 – 2,85) and (2,95 – 3,7). Finally, the abbreviation of MDF and PTB should be changed to the full name when they first appeared.

Authors answer: We have modified abstract according to another reviewers too.

Introduction: the writing of the current research review is fine. The research purpose is also well refined from the current literature review. However, some lately publications on vibration during machining are missed in the literature review. For instance, a paper on milling composites with spiral cutters (https://doi.org/10.1016/j.jmrt.2022.12.054) also reached the conclusion of  “Generally, the surface roughness decreased with increasing spindle speed…the increase of tool diameter [5-13]”. Another research (https://doi.org/10.1016/j.cja.2022.12.009) on titanium machining also mentioned the topic of “feed rate is the most dominating parameter affecting surface finish, whereas cutting speed is the major factor effecting tool vibration [20]”, which can be added to enrich your literature review.

Authors answer: We have enriched the literature review, thank you.

Table 1: what is the meaning of unit of “A÷B” ?  Does it mean from A to B?

Thank you for comment but we haven’t find this meaning in the manuscript

Improve the format of Fig 2, 3 and Table 3. Figures a and b need to have clear boundaries and cannot be mixed. Magnetic component should be pointed out in Fig. 2b.

Authors answer: We have corrected this.

The material of cutter should be given.

Thanks for your comment. This information is hard to find, it is know-how of manufacturer. We contacted manufacturer but he hasn't answered yet.

The labels should be given for the result of measurement on individual axes in Fig. 5.

We have replaced this figure.

Section 2: the units in “Varying feed rate of 2, 6, 10 m.s -1 was used, with spindle rotations 10 000 and 20 000 s -1” should be re-checked.

Thanks for you comments. We have corrected this of course.

Fig. 9, 10, 11: the label of “n 10 000 min -1,  n 20 000 min -1” should be moved to the blank position in the upper right corner in the figure.

We have corrected this.

Improve the conclusion.

Thanks for your comment. We have improved and enriched it.

Reviewer 3 Report

Please see the pdf file attached.

Author Response

My comments and observations are listed below: 

Research novelty and originality should be clearly stated in the abstract section and in the last paragraph of introduction section. In the abstract some quantitative results should be briefly given to show the prominence of this method against others found in the literature. Some quantitative results should be also clearly given in the conclusions section.

Authors answer:  We added some quantitative results in the conclusion section and stated research novelty in abstract and introduction. We shortened the abstract and made it more condensed as another reviewer suggested.

Conclusions section should be revised and written in a more scientific way. References in conclusions section should be avoided. It is better to have references in the rest of the manuscript. Conclusions should clearly mention the future perspectives of the work presented.

Authors answer: We added future perspectives of the results and methods used in the research to conclusion section.

The authors need to clearly state the limitations of their system and give some future perspectives related to new implementation directions with emphasis to their novel optimization system.

Authors answer: We have added this in the conclusion

In general, most of the figures should be enhanced as their quality is quite low. The authors should also increase the font to some figure captions as they cannot be easily reviewed.

Authors answer: We have replaced some figures and changed the font size.

The table summarizing the cutting parameters of experiment and tools used should be normally given as table rather than as figure to facilitate easy reading and review (Figure 4b).

Authors answer: Thanks for your suggestion, we have corrected this.

Figure 5 is too small to be reviewed and understood. This figure should be numbered to Figure 5a up to Figure 5d, in order to observe the axes of reference and results in terms of measurements of U(mV) and t(s). By the way, the first graph in Figure 5 does not have label for X‐axis (i.e. U [mV]). The whole figure should be revised.

Authors answer: We have corrected this.

The Language needs to be refined. Grammar and spelling errors should also be checked.

Authors answer: We made our best and checked the language.

Round 2

Reviewer 1 Report

The authors included all my suggestions in the revised version of the paper. I recommend the manuscript for publication.

Reviewer 2 Report

According to the reviewer’s opinion, the revised manuscript is well-structured and clear. The topic is interesting and falls within the aim of the journal. In addition, the results are well-presented and could be helpful to further develop the same topic. Therefore, the manuscript can be accepted for publication in the current form.

Reviewer 3 Report

The authors have made the required revisions according to the 

reviewers' comments. Therefore i think that the manuscript warrants publication.